

# Water Vapor Exchange between Atmospheric Boundary Layer and Free Troposphere over Eastern China: Seasonal Characteristics and ENSO Anomaly

Xipeng Jin[1], Xuhui Cai[2*], Xuesong Wang[2], Qianqian Huang[3], Yu Song[2], Ling Kang[2], Hongsheng Zhang[4], Tong Zhu[2]

[1]Collaborative Innovation Center of Atmospheric Environment and Equipment Technology, Jiangsu Key Laboratory of Atmospheric Environment Monitoring and Pollution Control, School of Environmental Science and Engineering, Nanjing University of Information Science & Technology, Nanjing 210044, China
[2]State Key Lab of Environmental Simulation and Pollution Control, College of Environmental Sciences and Engineering, Peking University, Beijing 100871, China
[3]Institute of Urban Meteorology, Beijing 100089, China
[4]Department of Atmospheric and Oceanic Sciences, School of Physics, Peking University, Beijing 100871, China

*Correspondence to*: Xuhui Cai (E-mail: xhcai@pku.edu.cn)

**Abstract.** This study develops a quantitative climatology of water vapor exchange between the
atmospheric boundary layer (ABL) and free troposphere (FT) over Eastern China. The
exchange flux is estimated for January, April, July, and October over 7 years (2011 and 2014-
2019) based on a water vapor budget equation using simulated meteorological data. The spatial
pattern of the ABL-FT water vapor exchange flux is closely related to the topographic
distribution. The seasonal variation shows that the water vapor exchange in the northern region
is downward in January and October with the flux being 37%-72% of the surface evaporation
to maintain the ABL moisture, while it is weak upward in April and July; the southern region
presents persistently water vapor output from the ABL to the FT, with the ratio of exchange
flux to surface evaporation increasing from 10% in January and October to 60%-80% in April
and July. Three physical processes determine the total water vapor exchange, among which the
ABL diurnal variation drives large magnitude exchange flux within the one-day cycle, but for
the net monthly mean flux, the vertical motion at the ABL top is the main contributor. The
anomaly of water vapor exchange in ENSO years illustrates triple antiphase distribution:
strengthening in the middle area and weakening in the north and south zones of Eastern China
in La Niña year, and vice versa in El Niño year. It agrees with the spatial pattern of anomalous
precipitation, implying the crucial role of ABL-FT water vapor exchange in atmospheric water
cycle.
**Keywords**: Water vapor; atmospheric boundary layer; free troposphere; vertical exchange



## 1 Introduction

Water vapor is a significant constituent in the atmosphere. It directly participates in fundamental physical processes, including cloud formation, precipitation, severe weather development and atmospheric circulation (Sodemann and Stohl, 2013; Wong et al., 2018; Wypych et al., 2018). Water vapor also affects important chemical reactions, such as providing OH radicals for gaseous photochemical transformations and serving as a medium in secondary aerosol formations (Pilinis et al., 1989; Tabazadeh 2000; Wu et al., 2019). Moreover, the radiation forcing of water vapor accounts for about 2/3 of the total natural greenhouse effect, which plays a vital role in climate feedback (Kiehl and Trenberth, 1997; Harries et al., 2008; Adebiyi et al., 2015).

The distribution of water vapor in the atmospheric system depends on its source and transport processes. In general, water vapor evaporates from the Earth's surface into the atmosphere. From the meridional and zonal view, it presents a transport trend from low latitude to high latitude and from ocean to land. The horizontal transport of water vapor has been widely discussed from multiple scales. Hemispheric-scale atmospheric rivers induce large excursions of high vertically integrated water vapor from the subtropics to high latitudes (Newell et al. 1992; Zhu and Newell 1998; Sodemann and Stohl, 2013). Synoptic-scale moisture flux convergence of extratropical cyclones explains the precipitations and cloud structures over the warm front and cold front (Boutle et al., 2010; Wong et al., 2018). Regional-scale transport processes are widely reported in many areas from water vapor advection and dynamical convergence (Zhou and Yu, 2005; Sun et al., 2010; Gvozdikova and Muller, 2021). However, these studies estimate vertically integrated water vapor through the atmospheric layer (usually from the surface to 300 hPa) or only focused on a certain altitude.

The water vapor vertical transport, especially within the troposphere, plays a key role in the atmospheric water cycle. All water vapor in the atmosphere originates from surface evaporation and is first confined in the atmospheric boundary layer (ABL, Boutle et al., 2010), which is defined as the lowest layer of the atmosphere influenced by the Earth's surface (Stull, 1988). The water vapor is turbulently mixed in the ABL, making it act as a reservoir. Actually, all water vapor entering and transporting meridionally and zonally in the free troposphere (FT) is initially exported through the ABL (Bailey et al., 2013). In other words, the water vapor exchange between the ABL and the FT is a prerequisite for its large scale transport and redistribution, as well as interaction with other constituents, in the upper atmosphere. Several studies indicate the importance of this key process on precipitation (Liu et al., 2020), cloud systems (Miura et al., 2007), tropical cyclone formation (Fritz and Wang, 2013), Madden–Julian oscillation (Hirota et al., 2018), West African Monsoon Jump (Hagos and Cook, 2007), and $O_3$ vertical distributions (Andrey et al., 2014). Therefore, it is of great significance to quantify the vertical exchange of water vapor between the ABL and FT.

However, the exchange between the ABL and FT is not straightforward, both for water vapor or air mass. Although the diurnal variation of the ABL depth allows air constituents to



be entrained into and left out of this layer within its variation range, the actual exchange
between ABL and FT is small on the time scale of more than one day due to the canceling
effect (Hov and Flatoy, 1997; Jin et al., 2021). The current studies on water vapor vertical
transport are mainly limited to complex terrain areas or special convective events. The
local/mesoscale circulation induced by orographic thermal and dynamic effects is considered
a key process for ABL ventilation (Kossmann et al., 1999; McKendry and Lundgren, 2000;
Dacre et al., 2007). Henne et al. (2005) found that there were elevated moisture layers in the
lower free troposphere in the lee of the Alps resulting from mountain venting. On average for
the 12-year period, ~30% of the water vapor of the Alpine boundary layer was vented to the
FT per hour during the daytime, which makes the total precipitable water within the elevated
moisture layer increase by ~1.3 mm. Another simulated study indicates that the moisture
exchange between the ABL and FT of mountainous topography can be about 3–4 times larger
than the amount of moisture evaporated from the surface in a specific ventilation event (Weigel
et al., 2007). The convective system, mainly mesoscale deep and shallow convection, is another
important factor leading to the vertical transport of water vapor. The isotope observations show
that the moisture transport pathways to the subtropical North Atlantic FT are linked to dry
convection processes over the African continent which effectively injects humidity from the
ABL to higher altitudes (Gonzalez et al., 2016; Dahinden et al., 2021). The water vapor budget
of the free troposphere of the maritime tropics shows that 20% of this source comes from
vertical convective transport (Sherwood, 1996). On the other hand, an idealized simulation
suggests that the warm conveyor belt ascent and shallow convective processes contributed
about equally to FT moisture (Boutle et al., 2010).
Though for these studies, general characteristics of long-term and wide-ranging ABL-FT
water vapor exchange are still unknown. These characteristics are closely bound up with the
atmospheric energy flow and the entire climate system, affecting clouds, precipitation and
radiation (Sodemann and Stohl, 2013; Wong et al., 2018; Wypych et al., 2018). For example,
small variations in upper atmospheric humidity over a large space-time scale can cause
systemic changes in the hydrological cycle and atmospheric circulation (Minschwaner and
Dessler, 2004; Sherwood et al., 2010; Allan, 2012). The climate state of water vapor vertical
exchange flux is critical for quantifying these specific effects. To fill this knowledge gap, the
present study calculates the water vapor exchange flux between the ABL and FT for 7 years
(2011&2014-2019) over Eastern China (20-42°N, 108-122°E) to establish the first quantitative
climatology view on this issue. The water vapor budget method is used, with the mesoscale
meteorological simulation providing input data. January, April, July, and October, respectively
representing winter, spring, summer, and autumn, are considered to discuss the seasonal
characteristics. Interannual differences are analyzed by investigating the impact of El Niño and
La Ninã events. On the basis of understanding the foundational features, we further attempt to
discuss the role of ABL-FT water vapor exchange playing in anomalous precipitation. The
arrangement of this paper is as follows. Data and methods are described in Section 2. The
seasonal characteristics and mechanism analysis, interannual variability and the relation with



anomalous precipitation are presented and discussed in Section 3. Finally, the findings of this
study are summarized in Section 4.

## 2 Data and methods

### 2.1 Observation data

Intensive ABL sounding data and routine surface meteorological data were used to evaluate
the performance of the Weather Research Forecast (WRF) model that provided the input data
for estimating exchange flux.
Intensive ABL sounding data: Two field experiments of intensive GPS (Global Positioning
System) sounding were carried out in Dezhou (37°16′ N, 116°43′ E), located in the middle of
the North China Plain (NCP) (Fig. 1b), from December 25, 2017, to January 24, 2018, and
from May 14 to June 14, 2018. Eight soundings were taken for each day, at 02:00, 05:00, 08:00,
11:00, 14:00, 17:00, 20:00 and 23:00 LT (i.e., UTC + 8). GPS radiosonde (Beijing Changzhi
Sci and Tech Co. Ltd., China) was used to obtain profiles of wind, temperature and humidity
with the ascending velocity being about 3-5 m s$^{-1}$. We eliminated the outliers from the original
data and averaged the profiles to an effective vertical resolution of 10 m. ABL heights were
determined with these data via the potential temperature profile method (Liu and Liang, 2010).
The reliability of the GPS sounding data has been systematically evaluated by Li et al. (2020)
and Jin et al. (2020).
Routine surface meteorological data: The hourly surface data of 137 routine observatories
distributed within the research domain were collected from the Chinese National
Meteorological Center. The dataset included information on wind speed and direction, air
temperature, relative humidity, air pressure, cloud coverage and precipitation, which was used
to evaluate the WRF simulation.

### 2.2 Three-dimensional meteorological simulation

The WRF model was conducted to provide three-dimensional meteorological data for the
estimation of ABL-FT water vapor exchange flux. Two nested domains (Fig. 1a) were
employed with horizontal grid resolutions of 30 and 10 km, respectively. The inner covered
Eastern China (20–42°N, 108–122°E), the main research region for the ABL-FT water vapor
exchange in the present work (Fig. 1b). Each domain had 37 vertical layers extending from the
surface to 100 hPa, with the vertical resolution being about 20-30 m below 200 m, increasing
to ~100 m at 750 m, ~250 m at 2000 m, ~350 m at 3000 m, ~600 m at 5000 m, ~900 m at 8000
m, ~1300 m at 11000 m and gradually enlarging to the top of the model. There were 24 layers
within 3 km to resolve the ABL and its upper FT. The meteorological initial and boundary
conditions were set using the US National Center for Environmental Prediction Final Analysis
(NCEP-FNL) dataset.



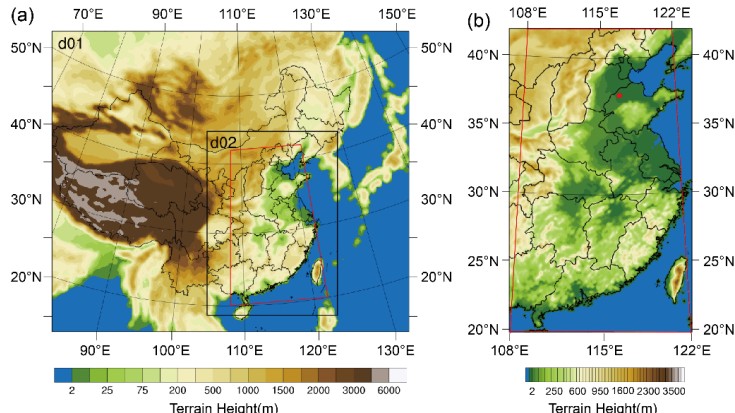


Figure 1. Geographical map of (a) the Weather Research and Forecast (WRF) model domains (d01 and d02) and (b) the amplified research domain (marked with red lines). The map uses the Lambert projection with the center meridians of 108°E in (a) and 115°E in (b). The red dot in (b) indicates the intensive GPS sounding observatory.

In order to adequately reproduce water vapor distribution and to correctly estimate the ABL-FT exchange flux, sensitivity simulations were carried out to choose reasonable physical parameterization schemes. We focused on the microphysical and cumulus parameterizations that are the most relevant to the moisture simulation. Microphysics in the model includes explicitly resolved water vapor, cloud and precipitation processes. Cumulus schemes are responsible for the sub-grid scale effects of convection and/or shallow clouds. Vertical fluxes due to unresolved updrafts and downdrafts are represented. Lin et al. scheme (Lin et al., 1983) and WRF Single-Moment 6-class (WSM6) scheme (Hong and Lim, 2006) in microphysics parameterization, and Grell-Devenyi (GD) ensemble scheme (Grell and Devenyi, 2002) and Kain-Fritsch (KF) scheme (Kain, 2004) in cumulus parameterization were compared, which were most commonly used in previous moisture simulation studies (Perez et al., 2010; Gonzalez et al., 2013; Jain and Kar, 2017; Qian et al., 2020). Other physics parameterization schemes used in this study included the Yonsei University PBL scheme (Hong et al., 2006), Noah land surface Model (Chen and Dudhia, 2001), Dudhia shortwave radiation scheme (Dudhia, 1989) and the rapid radiative transfer model (Mlawer et al., 1997) for longwave radiation. WRF simulations were initialized at 00 UTC on the day and there was a 12-h spin-up time before the start of each 48-h simulation. Domain outputs were sampled every hour for the whole simulation period (January, April, July, and October in 2011 and 2014-2019).

These schemes were evaluated by comparing simulated and observed specific humidity, temperature and wind speed, from their near-surface temporal evolution and vertical spatial structure. Another two key parameters, ABL height and precipitation were also concerned: the former directly affects the exchange flux results, and the latter characterizes the moisture budget. The hourly averages of model outputs were extracted from the grid points nearest to the observed sites for comparison. In the vertical direction, the modeled and sounding data




were simultaneously interpolated into the same height with 10 m intervals ranging from 50 m
to 3 km. Note that the ABL height was diagnosed with the potential temperature profile method
both for the simulations and for observation data, rather than using the default bulk Richardson
number method in the YSU scheme.

The results of sensitivity experiments showed that there were no appreciable differences
among various microphysical and cumulus parameterization schemes (Table S1 and S2). In
comparison, the combination of the WSM6 scheme and GD scheme performed better in
humidity simulation and was more effective in reproducing temperature, wind speed and ABL
height, especially in summer (Table S2). Therefore, these schemes were used in the present
study. Its simulation performance determines the reliability of the calculated flux results and
thus a comprehensive evaluation is provided here. The spatial-temporal evolutions of modeled
and observed meteorological fields are presented by the height-time cross sections of specific
humidity, potential temperature and wind speed, as well as the ABL height and precipitation
(Fig. 2). During the winter and summer months of the intensive GPS sounding, the simulated

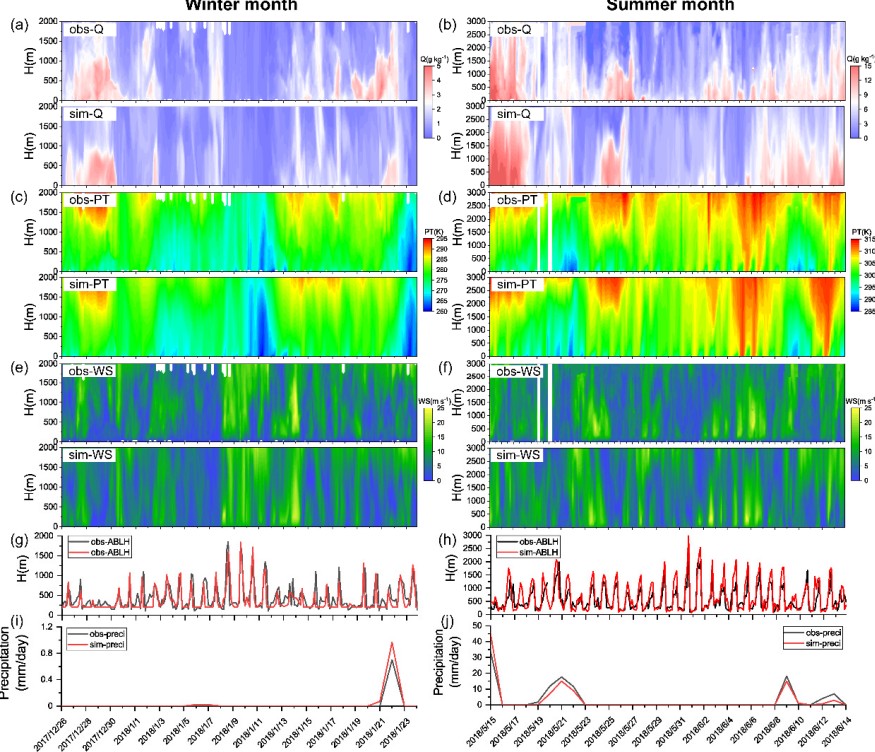


Figure 2. Observed and simulated time-height cross-sections of (a-b) specific humidity, (c-d) potential
temperature, (e-f) wind speed, and temporal evolution of (g-h) ABL height and (i-j) daily cumulative
precipitation at the Dezhou site (37.27°N, 116.72°E) during winter (from December 26, 2017, to
January 24, 2018) and summer (from May 15, 2018, to June 14, 2018) months of intensive GPS
sounding field experiment. The time resolution of sounding data in (a-h) is 3-hr.



atmospheric thermal and dynamic structures were comparable with observations. The
alternating between dry and wet atmospheric states (Fig. 2a-b), formation and decay of upper
temperature inversion (Fig. 2c-d), and vertical location and temporal transition of the strong
and weak wind layers (Fig. 2e-f) were successfully reproduced. Accordingly, a good
correlation between the simulated and observed ABL height was achieved, both in terms of
diurnal variation and synoptic evolution lasting several days (Fig. 2g-h). The correlation
coefficients were 0.71 and 0.84 during wintertime and summertime, respectively. It should be
mentioned that there was a slight discrepancy in the modeled ABL heights (mean biases are
about 70 m and 120 m in winter and summer), which may further affect the identification of
other parameters (such as the wind component) at the ABL top and lead to uncertainty in the
calculation results. This impact will be quantitatively analyzed in the discussion section.
Another concerned meteorological factor, the daily cumulative precipitation was also evaluated,
which showed a consistent evolution in observation and simulation (Fig. 2i-j) with correlation
coefficients as high as 0.99 and 0.91 ($p<0.05$) in winter and summer respectively,
demonstrating that the moisture budget is accurately captured by the WRF simulations. Overall,
the model showed the ability to capture the major variation of observed atmospheric thermal-
dynamical structures reasonably, which ensures the validity of the meteorological inputs for
the ABL-FT exchange flux calculation.
**2.3 ABL-FT water vapor exchange flux**
Similar to mass vertical exchange (Sinclair et al. 2010; Jin et al., 2021), the estimation of
ABL-FT water vapor exchange flux in this study was based on an ABL water vapor budget
equation established by Boutle et al. (2010):
$$\frac{\partial}{\partial t}\left(\int_0^h \rho q dz\right) = -\left(\frac{\partial}{\partial x}\int_0^h \rho q u\, dz + \frac{\partial}{\partial y}\int_0^h \rho q v\, dz\right) + (\rho q)_h\left(\frac{\partial h}{\partial t}\right)$$

$$-(\rho q)_h\left(\vec{U}\cdot\vec{n}\right)_h - \left(\rho\overline{w'q'}\right)_h + \left(\rho\overline{w'q'}\right)_0 + P, \tag{1}$$

where $\rho$ is air density, $q$ is water vapor mixing ratio, $h$ is the ABL height, $\vec{U}=(u,v,w)$ is
wind vector, $\vec{n}=(-\frac{\partial h}{\partial x}, -\frac{\partial h}{\partial y}, 1)$ is the unit normal vector perpendicular to the ABL top
surface, $w'$ and $q'$ are the fluctuation values of vertical velocity and water vapor content
respectively. $P$ is the precipitation. Subscripts $h$ and 0 indicate quantities at the ABL top and
the surface. The first term on the right side of Eq. (1) represents horizontal
convergence/divergence within the ABL, the second term indicates the local change in ABL
depth, the third term indicates vertical advection across the ABL top, the fourth and fifth terms
are turbulent transport at the ABL top and the surface respectively, and the last term indicates
the net precipitation falling through the ABL.
Denoting the water vapor vertical exchange flux between the ABL and FT as $F$ (positive
values represent upward transport), it can be further written as:
$$F = -((\rho q)_h\left(\frac{\partial h}{\partial t}\right) - (\rho q)_h\left(\vec{U}\cdot\vec{n}\right)_h - \left(\rho\overline{w'q'}\right)_h)$$



$$\approx -\left((\rho q)_h \frac{\partial h}{\partial t} + (\rho q)_h \left(u_h \frac{\partial h}{\partial x} + v_h \frac{\partial h}{\partial y}\right) - (\rho q)_h w_h\right). \qquad (2)$$

Since turbulent transport between the ABL and FT is typically related with dryer air that does
not affect the total moisture content, $\overline{(w'q')}_h$ is usually considered to be a negligible
contribution to the ABL-FT water vapor exchange flux (Boutle et al., 2010). Specifically, the
finite difference method was adopted for calculation with the time step being 1 hr, and the
horizontal dimensions of the model grid being 10 km. The ABL heights were obtained from
the hourly output of the WRF model. Other variables were extracted from the vertical level
closest to the top of the ABL. It is clear that the water vapor vertical exchange flux between
the ABL and FT is determined by i) the local temporal variation of ABL height, $\frac{\partial h}{\partial t}$, allowing
the water vapor entrained into the ABL or left in the upper atmosphere; ii) the spatial variation
of the ABL, making water vapor horizontally advected across an inclined ABL top; and iii) the
vertical advection motion, carrying water vapor downward/upward through the interface
between the ABL and FT. These three flux components are denoted as $F_{local}$, $F_{hadv}$, and $F_{vadv}$,
and their contributions and evolutions will be discussed in the following.
**3 Results and discussion**
The present study is based on a 7-year flux calculation. The years 2011&2014-2019 are
selected for analysis, which includes typical La Niña, El Niño, and neutral years (Marchukova
et al., 2020; You et al., 2021; Felix Correia Filho et al., 2021), and are considered to be valid
and concise datasets to reflect the characteristics of water vapor exchange between the ABL
and FT. Their climatic representativeness is demonstrated using a long-term historical dataset
provided by the fifth generation ECMWF (European Centre for Medium Range Weather
Forecasts) reanalysis. We compare the features of key meteorological elements during the
study period (2011&2014-2019) and over the past 30 years (1990-2019) by the Kolmogorov-
Smirnov test (K-S test) and histogram analysis. Temperature, three-dimension wind component,
specific humidity both near the surface and at the upper-level, as well as the ABL height and
precipitation are concerned. The K-S test indicates that there is no significant difference (with
a confidence level of 95%) between the 7-year sample period and the 30-year historical dataset
for these variables (Table S3). The histogram analysis further illustrates that their normalized
frequencies in the research samples are similar to that in the long-term historical data (Fig. S1).
Also, the annual variation of the two sets of data presents a high consistency, with similar mean
values and standard deviations (Fig. S2). The above analysis verifies that the 7-year samples
adopted in this study can represent the long-term climatology, and be promising to obtain
climatic features of water vapor exchange between the ABL and FT. The basic temporal and
spatial patterns, influencing mechanism, and relationship with ENSO and extreme precipitation
are revealed as follows.



### 3.1 Seasonal generality and variability

### 3.1.1 Spatial distribution

Figure 3 shows the spatial distribution of water vapor exchange flux between the ABL and FT in the research domain (20-42°N, 108-122°E, marked by red lines in Fig.1), averaged over all 7-year (2011, 2014-2019) for January, April, July, and October. It is obvious that the ABL-FT water vapor exchange in the north and south of the research domain is different, because they are affected by subtropical and temperate climates, respectively (Domroes and Peng, 1988; Zheng et al. 2013; Zhang et al., 2020). Therefore, the southern (20-32°N, 108-122°E) and northern (32-42°N, 108-122°E) regions are divided for analysis (the boundary marked in Fig. 3). Water vapor exchange is more active in the southern region with more pronounced spatial variability, and tends to output from the ABL. In the northern region, vertical exchange fluxes and spatial differences are relatively small. From another perspective, the vertical exchange of water vapor is closely related to the topographic distribution (Fig. 1b), which is manifested as strong exchange activities usually occurring around mountainous or coastal areas, both in the northern and southern regions. This feature is similar to the spatial pattern of the air mass exchange flux between the ABL and the FT indicated by Jin et al. (2021). It is the result of the dynamical interaction of topography on the synoptic system, and thermal property difference over the heterogeneous underlying surface (Kossmann et al., 1999; Dacre et al., 2007; Jin et al., 2021). These phenomena will be detailedly explained in the mechanism analysis in Sect. 3b.

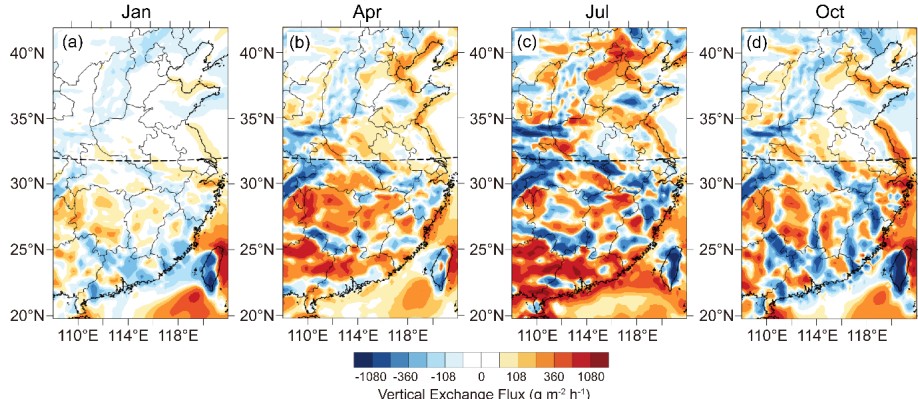

Figure 3. Spatial distribution of ABL-FT water vapor exchange fluxes in Eastern China, averaged over 7-year for (a) January, (b) April, (c) July, and (d) October. Black dashed lines mark the boundary between the northern (32-42°N, 108-122°E) and southern (20-32°N, 108-122°E) regions. Positive and negative fluxes (warm and cool colors) represent water vapor upward and downward transport at the ABL and FT interface.





### 3.1.2 Seasonal difference

Corresponding to Fig. 3, the spatial means of ABL-FT water vapor exchange flux and their seasonal evolutions for northern and southern regions are shown in Fig. 4. They are obtained by grid averaging in the ranges of 32-42°N, 108-122°E and 20-32°N, 108-122°E, respectively. Obviously, the exchange flux varies from season to season in both regions. For the northern region, winter and autumn (represented by January and October, respectively) are characterized by water vapor transport downward from the FT into the ABL, with the spatial mean fluxes of $-15.6$ and $-18.8$ g m$^{-2}$ h$^{-1}$ (1 g m$^{-2}$ h$^{-1}$ $=10^{-3}$ mm h$^{-1}$) and the standard deviation of 3.6 and 8.6 g m$^{-2}$ h$^{-1}$ over 7 years. While in spring and summer (represented by April and July, respectively), the northern region as a whole presents an upward export of water vapor from the ABL to the FT, with the regional mean fluxes being 6.4 and 11.9 g m$^{-2}$ h$^{-1}$. They are characterized by more significant inter-annual variations than the exchange fluxes in the cold seasons. In the southern region, the water vapor vertical exchange is featured with ABL output in all seasons, with a winter minimum and a summer maximum. The mean upward fluxes vary greatly, showing one order of magnitude greater in April and July (99.1 and 115.51 g m$^{-2}$ h$^{-1}$) than in January and October (9.6 and 16.7 g m$^{-2}$ h$^{-1}$), accompanied by the larger standard deviation (50.4 and 68.4 g m$^{-2}$ h$^{-1}$). The notable interannual variability in the warm season may be related to the ENSO phenomenon, which will be discussed in the following section.

In order to better understand the magnitude of water vapor exchange between the ABL and FT, we compare the transport flux with the surface evaporation rate (Table 1). It indicates the "emission intensity" of water vapor from the surface, which varies in different regions and seasons. The surface evaporation rates in the northern and southern regions have maximums in summer (122.4 g m$^{-2}$ h$^{-1}$ and 194.4 g m$^{-2}$ h$^{-1}$) and minimums in winter (21.6 g m$^{-2}$ h$^{-1}$ and 108.0 g m$^{-2}$ h$^{-1}$). Obviously, the evaporation in the north is weaker than that in the south, especially in winter, it is only one-fifth of that in summer. Consequently, for the northern region, during the cold seasons with the dry land surface, the ABL-FT water vapor exchange is downward and the input flux is 37%-72% of the surface evaporation rate. Although the specific humidity decreases with height, counter-gradient transport still occurs reasonably because the ABL-FT exchange is a typically non-local mixing process (Stull 1988; van Dop and Verver, 2001; Ghannam et al., 2017). This suggests the ABL is a net moisture sink of upper layer FT air, which plays a role in maintaining water vapor within this layer. As surface evaporation intensifies in the warm months, water vapor is exported from the ABL in April and July, and the upward flux accounts for 10% of the evaporation rate. In the southern region with relatively strong evaporation, the ABL water vapor is always transported upward to the FT. The output flux is about 10% of the evaporation rate in January and October, and this ratio is as high as 60%-80% in April and July, indicating that the ABL acts as an effective water vapor source to the upper atmosphere.



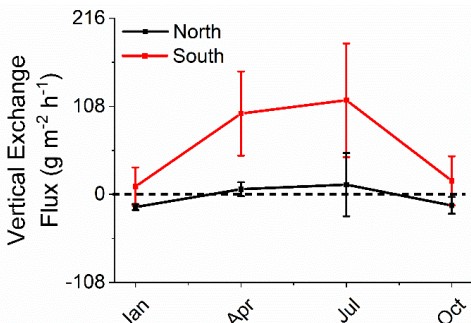

Figure 4. Seasonal variation of average ABL-FT water vapor exchange fluxes and their standard deviations over the northern region (32-42°N, 108-122°E) and southern region (20-32°N, 108-122°E) during 7 years. Positive and negative fluxes represent water vapor upward and downward transport between the ABL and FT.

Table1. Comparison of ABL-FT water vapor exchange flux (g m$^{-2}$ h$^{-1}$, positive for upward, negative for downward) and surface evaporation rate (g m$^{-2}$ h$^{-1}$, positive for upward) in the northern and southern regions.

| Region | Process | Jan | Apr | Jul | Oct |
|---|---|---|---|---|---|
| North | ABL-FT exchange | −15.6 | 6.4 | 11.9 | −18.8 |
| | Surface evaporation | 21.6 | 61.2 | 122.4 | 50.4 |
| South | ABL-FT exchange | 9.6 | 99.1 | 115.5 | 16.7 |
| | Surface evaporation | 108.0 | 115.2 | 194.4 | 144.0 |

**3.2 Main influential mechanism**

As shown in Eq. (2), three physical terms contribute to the total ABL-FT exchange, i.e., the local temporal variation of ABL height ($F_{local}$), the horizontal advection across the spatial inclined ABL top ($F_{hadv}$), and the vertical motion through the ABL-FT interface ($F_{vadv}$). It is of interest to clarify the specific effects of these factors on water vapor vertical exchange and their seasonal characteristics. Results of the monthly mean and diurnal cycle over the 7 years are presented below respectively.

The monthly mean results show that the term $F_{vadv}$ is the most significant to total ABL-FT moisture exchange flux (Fig. 5, green bar). In the northern region, this term produces persistent downward flux (−19.5~−44.7 g m$^{-2}$ h$^{-1}$, Fig. 5a), which substantially offsets the upward flux caused by the other two terms, so that the ABL water vapor presents net input during cold months (i.e., January and October) and weak output in warm seasons (i.e., April and July). For the southern region, it induces small downward fluxes in January and October (−18.6 and −5.5 g m$^{-2}$ h$^{-1}$) while large upward flux in April and July (60.7 and 68.6 g m$^{-2}$ h$^{-1}$), which results in the total water vapor exchange as weak and strong output from the ABL during cold and warm months, respectively (Fig. 5b).

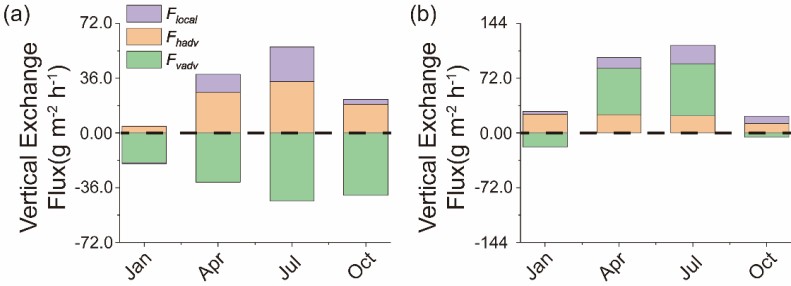

Figure 5. Contributions of three components ($F_{local}$, $F_{hadv}$, and $F_{vadv}$) to the total ABL-FT water vapor exchange flux. Results are spatial mean over the (a) northern (32-42°N, 108-122°E) and (b) southern (20-32°N, 108-122°E) regions of Eastern China respectively. $F_{local}$: local temporal variation of ABL height (purple bar); $F_{hadv}$: advection across the spatial inclined ABL top (yellow bar); $F_{vadv}$: vertical motion through the ABL-FT interface (green bar). Positive and negative fluxes represent water vapor upward and downward transport between the ABL and FT.

The upward/downward transport of water vapor caused by the term $F_{vadv}$ depends on the direction of the vertical motion. The spatial distributions of the vertical velocity are presented in Fig. 6, accompanied by horizontal wind fields at the ABL top, as well as terrain heights. The upward motions usually occur on the windward of the mountains, while the descending velocities appear on the leeward side, in each season. This is attributed to the dynamic forcing of the terrain on seasonal mean winds. Due to the alternation of winter and summer monsoons throughout the year, the vertical motion pattern varies accordingly in four representative months (Fig. 6a-d). In the winter, the Siberian high invades from the northwest and forms strong northerly winds (Fig. 6e). In the northern region, the prevailing northwest airflows overcome the obstruction of Taihang Mountain and intensely descend on its leeward side (Fig. 6a). As the air migrates south, the dominant airflow deflects northeasterly (Fig. 6e), and the vertical motion manifests more upward velocities in front of the major mountainous region, and more downward velocities behind these mountains (Fig. 6a). During the summer, southerly air flows dominate eastern China and gradually weaken from south to north (Fig. 6g). The southern region is characterized by obvious forced uplift on the windward side of the major mountains (Fig. 6c). The onshore airflow convergence of the prevailing southerly winds in coastal areas also produces upward motions (Fig. 6c). These factors are conducive to the vertical output of ABL water vapor in the southern region during warm months. The northern region is less invaded by the summer monsoon: only the eastern part of the NCP is affected by southerly winds to induce upward motion in the piedmont, while the western part is still dominated by westerly winds leading to systematic subsidence (Fig. 6c, g). The general patterns of vertical velocity fields provide an explanation for the water vapor exchange fluxes caused by the term $F_{vadv}$. It is noticed that, although the ABL-FT water vapor exchange fluxes in Fig. 3 are averages over 7 years, there still exists obvious spatial heterogeneity. Smooth variations in both the mean wind field (Fig. 6e-h) and mean ABL height (Fig.S5) indicate these two factors are not related to the flux heterogeneity. But there indeed exists discontinuous

structures in the vertical velocity fields at the ABL top (Fig.6a-d), which is significant to water
vapor exchange flux. There can be smaller-scale secondary vertical motion being stimulated
when prevailing airflows encounter diverse terrains (Fig. S4). Multiscale dynamical
interactions between complex terrain and synoptic processes should be of great significance to
the water vapor exchange between the ABL and FT.

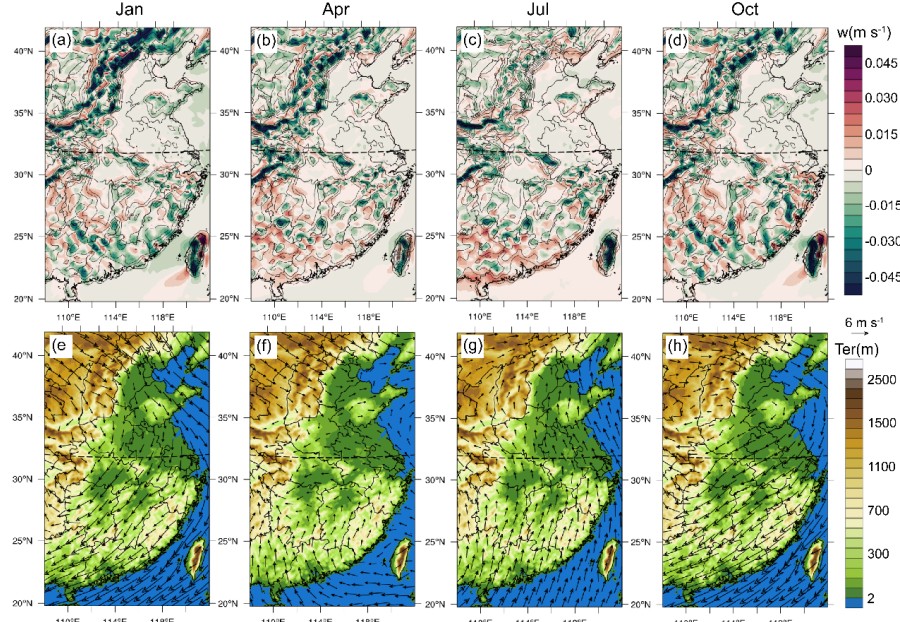


Figure 6. Spatial distribution of (a-d) vertical velocities at the ABL top and (e-h) terrain height
superposed with horizontal wind vectors averaged over 7-year for January, April, July, and October.
Positive values represent upward motions and the contours in (a-d) represent the terrain height. Black
dashed lines mark the boundary between the northern (32-42°N, 108-122°E) and southern (20-32°N,
108-122°E) regions.
The horizontal advection term $F_{hadv}$ tends to allow water vapor to be out of the ABL and
the magnitude increases in spring and summer (Fig. 5, yellow bar). This water vapor exchange
component mainly occurs in the mountain-plain transition zone and the land-ocean boundary
(Fig. S3e-h), where the ABL is unevenly distributed due to the heterogeneous surface
properties (Fig. S5). During the warm season, the thermal difference is more obvious with the
solar radiation strengthening and thereby with larger spatial variation of the ABL, especially
in the northern region. This explains the seasonal variation of the water vapor exchange flux
caused by the term $F_{hadv}$.
The temporal ABL height variation term $F_{local}$ contributes relatively less to the total water
vapor exchange (Fig. 5, purple bar). Noticeably, this average flux component is positive, being
negligible in autumn and winter (0.7~3.3 g m$^{-2}$ h$^{-1}$), but becoming relatively pronounced in
spring and summer (12.0~24.5 g m$^{-2}$ h$^{-1}$). This is inconsistent with the air mass exchange



between the ABL and FT, in which the monthly average flux caused by this term is always
insignificant because the ABL entrainment and detrainment of the air mass cancel out each
other in a diurnal cycle (Jin et al., 2021). To understand more details of the term $F_{local}$ in the
ABL-FT water vapor exchange, the mean diurnal variation of the exchange flux is derived and
shown in Fig. 7.

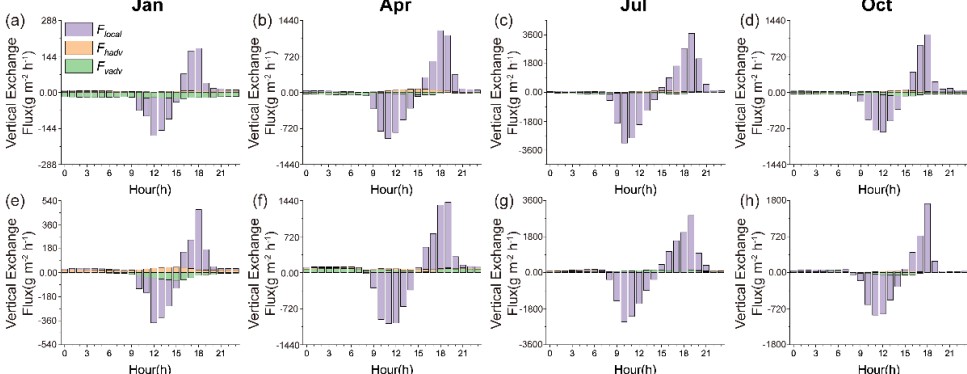

Figure 7. Diurnal variation of the three exchange flux components ($F_{local}$, $F_{hadv}$, and $F_{vadv}$) over the
(a-d) northern region (32-42°N, 108-122°E) and (e-f) southern region (20-32°N, 108-122°E) averaged
for (a, e) January, (b, f) April, (c, g) July, and (d, h) October. $F_{local}$: local temporal variation of ABL
height (purple bar); $F_{hadv}$: advection across the spatial inclined ABL top (yellow bar); $F_{vadv}$: vertical
motion through the ABL-FT interface (green bar). Positive and negative fluxes represent water vapor
upward and downward transport between the ABL and FT.

At a first sight of the daily cycle, $F_{local}$ is the absolutely dominant term in all seasons and
both northern and southern regions (Fig. 7, purple bar), corresponding to the diurnal variation
of the ABL height (shown in Fig. S6). When the unstable ABL develops in the morning, the
water vapor in the residual layer is entrained into the ABL; while as the daytime ABL collapses
in the later afternoon, a large part of water vapor is left aloft the newly formed stable ABL.
Note that, unlike the air mass exchange at the ABL top, the water vapor entrained (input) flux
is less than the output flux, especially in spring and summer. This difference can be attributed
to the fact that the surface is, in general, a continuous evaporation source throughout a diurnal
cycle. Turbulent mixing brings water vapor upward in the ABL depth, and forms a net upward
flux across the ABL top. This is also the reason why a larger magnitude of $F_{local}$ exists in the
warm seasons when there is more strong surface evaporation. Although the ABL temporal
variation term $F_{local}$ dominates the diurnal variation of the total ABL-FT moisture exchange
flux, it contributes only a weak net output of water vapor in a monthly average flux, in
comparison with the vertical motion term $F_{vadv}$, as mentioned above.
**3.3 Interannual variability and its relation with anomalous precipitation**

A climatic mean of the ABL-FT water vapor exchange over eastern China is presented
above. Critically linked to the atmospheric water cycle, the exchange flux and its interannual



variation are of great interest. It is well known that the atmospheric water cycle is significantly
affected by El Niño and southern oscillation (ENSO), which is a joint phenomenon of the ocean
and the atmosphere appearing as a recurring anomaly of the sea surface temperatures in the
tropical Pacific and a seesaw of sea level pressure anomalies between Tahiti and Darwin. The
El Niño (warm phase) and La Niña (cold phase) are the two extremes of ENSO (Walker and
Bliss, 1932, 1937; Kousky et al., 1984; Wolter and Timlin, 2011). Considerable work has been
conducted on the relationship between ENSO and wet and dry variability, water vapor
horizontal transport, and precipitation events (Diaz, 2000; Knippertz and Wernli, 2010; Felix
Correia Filho et al., 2021). However, little is known about the ABL-FT water vapor exchange
during ENSO events. Here we take July as the research object, the month with the largest
variability (shown in Fig. 4), to investigate the interannual difference of ABL-FT water vapor
exchange fluxes affected by the ENSO phenomenon.

The correlation between the water vapor exchange flux anomalies and the Niño-3.4 index
during the study period (2011&2014-2019) is quantitatively calculated. The former (anomaly
or variability) is derived from the difference of each year with the 7-year average, and the latter
is obtained from the website of https://psl.noaa.gov/gcos_wgsp/Timeseries/Nino34/,
representing the average equatorial sea surface temperature across the Pacific from about the
dateline to the South American coast (5°N-5°S, 170°W-120°W), which is the most commonly
used indices to define El Niño and La Niña event. The statistical result shows that there is a
significant correlation between the two factors (with a confidence level of 95%). A negative-
positive-negative triple distribution is presented in the correlation map (Fig. 8). On this basis,
the sensitive areas are identified, in which the water vapor exchange fluxes are further analyzed.
For the central region (28-35°N, 108-122°E) with obvious positive correlation, the mean
vertical output flux of water vapor is enhanced by about 57.6~151.2 g m$^{-2}$ h$^{-1}$ in La Niña years

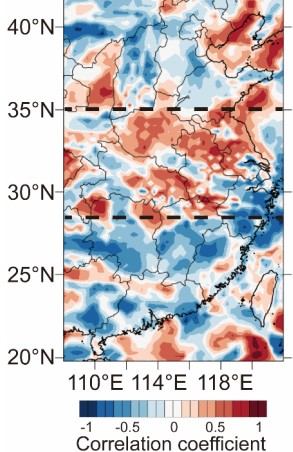


Figure 8. Spatial distribution of correlation coefficient between the water vapor exchange flux
anomalies and Niño-3.4 index in July for 7 years. The black dashed lines indicate the tripole distribution.



(2011 and 2016, blue boxes in Fig. 9a), and vice versa in El Niño years (2015 and 2019, red
boxes in Fig. 9a), and the flux anomalies are close to 0 in neutral years (2014, 2017, and 2018,
black boxes in Fig. 9a). In south and north areas with negative correlation coefficients, it is
reversed. That is, the ABL moisture ventilation flux weakens 79.2~140.4 g m$^{-2}$ h$^{-1}$ in La Niña
years and increases 108~194 g m$^{-2}$ h$^{-1}$ in El Niño years (figure not shown). This provides an
explanation for the interannual variation of the water vapor exchange flux mentioned in Sect.
3a. Further analysis of the three physical processes causing vertical transport suggests that the
ENSO phenomenon affects the water vapor exchange mainly by modifying the vertical motion
patterns at the ABL top, which may fundamentally change other weather processes in this
region, e.g., the distribution of precipitation.

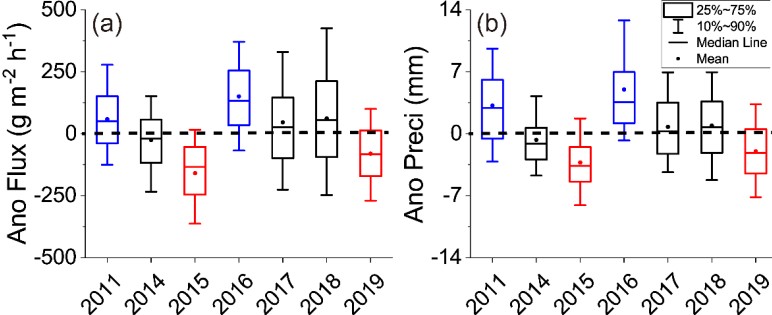


Figure 9. Anomalies of (a) water vapor exchange flux and (b) precipitation in July over the central
region (28-35°N, 108-122°E, indicated in Fig. 8) during 2011&2014-2019. Blue, red and black indicate
La Niña years, El Niño years and neutral years, respectively. Upper and lower sides of the box are the
75th and 25th percentile, and whiskers are the 90th and 10th percentile. Hollow squares and black lines
in the box are mean and median.
Previous observation climatological studies have indicated that the summer precipitation
anomalies in La Niña/ El Niño years are characterized by a tripolar distribution over eastern
China (Wang et al., 2020), similar to water vapor exchange flux anomalies revealed in this
work. It is of interest to investigate the relationship between water vapor vertical exchange and
precipitation under the influence of ENSO. Taking the central region (28-35°N, 108-122°E) as
an example, the precipitation anomalies present a good correspondence with the variations of
the ABL-FT water vapor exchange flux (Fig. 9). Specifically, precipitation increases (decreases)
about 3.2-6.9 mm (2.8-3.5 mm) when the vertical output of water vapor intensifies (weakens)
in La Niña (El Niño) years. That is, enhanced water vapor output flux from the ABL to the FT
tends to produce increased precipitation and vice versa. These results imply that, upper layer
FT water vapor supplement from the ABL can also be a significant factor to change regional
precipitation, in addition to horizontal transport.
It should be stated that the above results are preliminary and rough, due to the limitations
of the sample. The response of the ABL-FT water vapor exchange to ENSO, and its impact on
precipitation, are complicated. The isolated patches in Fig. 8, as well as the box and whisker in



Fig. 9 (being 75th-25th and 90th-10th percentile of the flux/precipitation anomalies), reflect
the complex spatial variability over the research domain, which is not thoroughly analyzed in
the current work. Nevertheless, this general result points to an association among ABL-FT
water vapor exchange, ENSO, and extreme precipitation, which should be paid more attention
to in future research.
**3.4 Discussion**
The present results are based on numerical simulations. Although reasonable
parameterization schemes are chosen according to sensitivity experiments, and the model
performance is also evaluated by observational data, there are inevitable uncertainties in the
modeled meteorological fields, which may directly affect the estimate of ABL-FT water vapor
exchange flux. For example, the difference between the simulated ABL height and the observed
value (~70 m and 120 m in winter and summer) brings ~30% uncertainty to the acquisition of
vertical velocity at this level, which may affect the accuracy of the flux results in a similar
magnitude. In addition, ignoring the turbulence term in this study may also reduce the accuracy
of the results. Nevertheless, this work presents a general view of long-term and large-scale
ABL-FT water vapor exchange over Eastern China.
The water vapor exchange in the climatological sense presents a significant regional
division of north and south China, due to their quite distinct climatic features. In addition to
this general pattern, the spatial heterogeneity associated with the topographic distribution is
also noteworthy. We try to sort out the vertical exchange fluxes of water vapor over the ocean,
plain and mountain, roughly by the altitude below 0m, between 0-200m and greater than 200m.
The statistical results show that the ocean and plain are characterized by the upward output of
water vapor from the ABL, while the transport is downward over the mountainous region. To
further discuss the causes of these results is of interest, but it is quite beyond our objectives in
this preliminary work. The present results indicate the importance of interaction between
synoptic airflow and complex terrain in the long run. It suggests that topographic ventilation is
not only caused by mesoscale circulations such as daytime upslope winds/sea breezes around
mountains/coasts (Henne et al., 2004; Weigel et al., 2007) or convective activities on a
relatively small scale or a specific time (Gonzalez et al., 2016; Dahinden et al., 2021).
Dynamical forcing of terrain on seasonal airflow or synoptic winds is more essential, which
induces vertical motion and leads to systematic water vapor exchange.
Moreover, the climatology of water vapor exchange flux between ABL and FT provides a
quantitative background for investigating weather processes, radiation feedback and climate
changes. Water vapor entering the FT may provide more latent heat to the energy flows and
further affect synoptic systems. It is also involved in the radiative budget to influence climate.
Previous model simulations and observations indicate that small yet systematic changes in the
humidity of upper atmosphere modulate the magnitude of the hydrological cycle and radiative
feedback, including clouds and precipitation (Minschwaner and Dessler, 2004; Sherwood et al.,
2010; Allan, 2012). Our results also demonstrate a notable relation between precipitation





anomalies and ABL-FT water vapor exchange patterns. Based on the quantitative results in this
study, the specific role of ABL - FT water vapor exchange in Earth's energy flows and climate
system might be studied further in the future.
**4 Summary**
In this study, we developed a climatology of water vapor exchange flux between the ABL
and FT, based on 7-year meteorological modeling data. The ABL water vapor conservation
method was used to estimate the vertical exchange flux across the ABL-FT interface. Spatial
distribution and seasonal characteristics of the water vapor exchange were presented, and the
interannual variability was simply discussed through their variations in ENSO events. Three
influential mechanisms of the water vapor exchange between ABL and FT were also analyzed.
The major findings of this work are the following:
The spatial pattern of the ABL-FT water vapor exchange flux was closely related to the
topographic distribution in each seasonal representative month (January, April, July and
October), with strong exchange activities occurring over mountainous areas and coastal areas.
In the northern region (32-42°N, 108-122°E), the winter and autumn months (January and
October) were characterized by the net downward flux of water vapor ($-15.6$ and $-18.8$ g m$^{-2}$
$^{2}$ h$^{-1}$), being 37%-72% of the surface evaporation. The water vapor downward transport from
FT was another source for ABL moisture maintenance in these drier and colder seasons. During
the spring and summer months (April and July), the water vapor was exported from the ABL
with the regional average flux of 6.4 and 11.9 g m$^{-2}$ h$^{-1}$. In the southern region (32-42°N, 108-
122°E), the water vapor vertical exchange at the ABL top was persistently upward to the FT.
And the flux accounted for about 10% of the surface evaporation rate in autumn and winter
(9.6 and 16.7 g m$^{-2}$ h$^{-1}$), and increased to 60%~80% during warm seasons of spring and summer
(115.5 and 99.1 g m$^{-2}$ h$^{-1}$). Clearly, the ABL acted as a channel to transport surface moisture to
the FT, particularly in the southern region during summer.
Three physical terms determined the total ABL-FT exchange of water vapor, i.e., the
diurnal variation of ABL height, the air advection across the inclined ABL top, and the vertical
motion through the ABL-FT interface. The respective contributions of these three terms were
revealed. The first term showed prevailing diurnal variation, but achieved only a small upward
water vapor transport in the average of longer than a one-day cycle. The second term tended to
cause the water vapor output from the ABL, especially in spring and summer. In a view of the
monthly average, the third term was the most prominent, which played a determinative role in
contributing total downward flux in the northern cold months and the total upward flux in the
southern warm months.
Interannual variability of ABL-FT water vapor exchange was demonstrated by the results
in ENSO event years. The exchange flux was strengthened in the middle zone and weakened
in the north and south of Eastern China in La Niña year (vice versa in El Niño year), presenting
as a triple anti-phase distribution. Moreover, the exchange flux variation illustrated good



correspondence with precipitation anomalies, shown as precipitation increasing accompanied
by stronger water vapor output in the middle area and precipitation decreasing in north and
south of Eastern China with the less upward flux of moisture in La Niña years, while the El
Niño years are opposite. This phenomenological analysis indicates a significant relation
between regional ABL-FT water vapor exchange and precipitation anomalies.
This work quantitatively reveals the climatological basic state of ABL-FT water vapor
exchange flux over Eastern China and demonstrates its significance in regulating the
atmospheric water cycle. The results may provide new insights for understanding and
predicting precipitation anomalies on large scales.
**Data availability**
The data in this study are available from the corresponding author (xhcai@pku.edu.cn).
**Author contribution**
XHC and XPJ designed the research. LK and HSZ collected the data. XPJ performed the
simulations and wrote the paper. XHC reviewed and commented on the paper. QQH, YS, XSW
and TZ participated in the discussion of the article.
**Competing interests**
The authors declare that they have no conflict of interest.
**Acknowledgements**
This work was supported by National Key Research and Development Program of China
(2018YFC0213204).

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
