# Peer review of "Water Vapour Exchange between Atmospheric Boundary Layer and Free Troposphere over Eastern China: Seasonal Characteristics and ENSO Anomaly"

_EGUsphere, 2023_

## Author Comment (AC1)

**Manuscript:** Water Vapour Exchange between Atmospheric Boundary Layer and Free Troposphere over eastern China: Seasonal Characteristics and ENSO Anomaly (egusphere-2023-1639)

**Response to Reviewer #1:**

**Summary**

*This is an interesting and well written paper, discussing water vapour exchange between the boundary layer and free troposphere over China, using budgeting techniques. My main concern is whether the results and conclusions presented are significant enough and of sufficient general interest. In a revised version of the manuscript I would encourage the authors to really focus on ensuring the abstract and conclusions highlight what they feel are the key new findings presented by their work, and what the general interest of these are (i.e. a wider ranging interest than just local meteorology over China). This may require further work on some aspects of the paper that were tantalising but lacked depth. The detailed comments are listed below.*

**Response:** We thank the reviewer for the positive evaluation and appreciate these valuable suggestions. We have conducted further analysis on some topics of concern, such as explaining why the correlations exist between water vapour vertical exchange and ENSO, and discussing the mechanism responsible for terrain dependence of exchange flux distribution. Meanwhile, the key findings and general interest of this work have been highlighted in the abstract and conclusion. Detailed responses are presented in the following respective terms. The original comments are in *blue and italic*, our replies are in normal font. Bracketed numbers are used for referee comments (e.g., *[R1.1]*).

**Major comments**

*[R1.1] Sect 3.3 - do the authors have a suggestion why the water vapour exchange changes in the way it does due to ENSO - what is the mechanism for this relationship? i.e. what components and inputs of the water budget are altered, and why? I think this is one of the key general interest parts of the paper, but it requires more in-depth discussion than merely presenting correlations, given you've gone to the effort of devising the budgeting tools that allow you to answer the questions of why the correlations exist.*

**Response:** We thank the reviewer very much for this comment. The reason for the change of water vapour exchange with ENSO is discussed. Three exchange components

$(F_{local}, F_{hadv}, F_{vadv})$ are analysed to identify key influencing factors. As shown in Fig. R1, the anomaly of $F_{vadv}$ has a good correspondence with the relevant pattern, suggesting that the ENSO affects the ABL-FT water vapour exchange mainly by modifying the vertical motions and water vapour contents at the ABL top. Taking the central region (with the most significant correlation) as an example, the upward vertical velocity strengthened and the water vapour mixing ratio increased in La Niña year (Fig. R2a-b), which together led to an enhancement of water vapour output from the ABL, while the opposite phenomenon occurred in El Niño year (Fig. R2c-d). This alteration mode is associated with stronger (weaker) southerly wind bringing more (less) water vapour and strengthened (weakened) horizontal convergence facilitating (reducing) vertical uplift in La Niña (El Niño) years, which are the results of ENSO's influence on the East Asian monsoon, according to previous studies (Zhou et al. 2012; Xue et al., 2015; Gao et al., 2018). The role of ENSO in regulating the monsoon has been widely reported, so it can be inferred that the further impact on water vapour vertical exchange is not only a local feature in eastern China, but rather a general phenomenon.

[Figure]

Figure R1. Spatial distribution of ABL-FT water vapour exchange flux anomalies for three components ($F_{vadv}, F_{hadv}, F_{local}$) in July of (a-c) 2016 (La Niña year) and (d-e) 2015 (El Niño year). $F_{vadv}$: vertical motion through the ABL-FT interface; $F_{hadv}$: advection across the spatial inclined ABL top; $F_{local}$: local temporal variation of ABL height. The black dashed lines indicate the triple distribution from north to south.

[Figure]

Figure R2. Spatial distribution of anomalies of vertical velocities (left) and water vapour mixing ratio (right) at the ABL top in July of (a-b) 2016 (La Niña year) and (c-d) 2015 (El Niño year).

The above analysis has been added in the revised manuscript at L490-502, and the corresponding figures are presented as new Fig. S7 in the supplementary material and new Fig. 10 in the revised manuscript. For the reviewer's convenience, these revisions are displayed below.

"In order to elucidate why the water vapour vertical exchange flux varies with ENSO, we further analyse three exchange components anomalies in El Niño and La Niña years (Fig. S7). Among them, the term $F_{vadv}$ presents the most obvious correspondence with the correlation pattern (Fig.8), demonstrating that vertical motion and water vapour content at the ABL top are crucial influencing factors. We select the central region (with the most significant correlation) for detailed analysis. As shown in Fig.10, in La Niña year (represented by 2016), the upward vertical velocity strengthened and the water vapour mixing ratio increased in the central area, while the opposite trend was observed in El Niño year (represented by 2015). This phenomenon is attributed to the stronger East Asian monsoon that brings more water vapour from the south and facilitates convergence to uplift during the cold phase period of ENSO, while in the warm phase, the weaker southerly wind reduces water vapour transport and is not conducive to convergence within the ABL (Zhou et al. 2012; Xue et al., 2015; Gao et al., 2018), which explains the increase or decrease of ABL water vapour output affected by ENSO."

*[R1.2] L507 - I accept that a complete analysis here is beyond the scope of this study, but I think the paper would be strengthened by at least providing some discussion of the mechanisms responsible for the water vapour transport results presented.*

**Response:** According to this suggestion, the mechanisms responsible for the water vapour exchange features over the plain, mountain and ocean are discussed. The topographic dependence of vertical exchange flux is due to the interaction between complex terrain and synoptic airflows. When the prevailing wind is blocked by the mountains, it is forced up on the windward side and then descends on the lee side. After reaching the plain, it slows down to converge, thus inducing upward motion. This explains the water vapour output from the ABL in the plain region. Over the mountains, the subsidence on the lee side is stronger than the lifting on the windward side, so the net water vapour transport is downward. As for the ocean area, the ABL water vapour output is attributed to the horizontal advection across the inclined ABL top, which occurs intensively over the nearshore waters. Although the above discussion is preliminary, it points to the terrain-dependent feature of water vapour vertical exchange and the importance of the interaction between mountain/sea and synoptic airflow. These findings provide a useful reference for understanding the ABL-FT water vapour exchange in other mountainous/coastal regions of the world.

The discussion has been added at L539-559 in the revised manuscript, as follows:

"The statistical results show that the ocean and plain are characterized by the upward output of water vapour from the ABL, while the mountainous regions are dominated by downward transport. This mode reflects the important role of the complex terrain in causing ABL-FT vertical exchange. As described in Sect. 3.2, the prevailing airflow is obstructed by the mountains to forcingly ascend on the windward and densely descend on the leeward slope, then it decelerates and converges to induce upward motion when reaching the plain area. This vertical motion pattern makes the water vapour upward export from the ABL in the plain, and downward transport in mountainous areas due to the intensity and effect of the leeward side subsidence being larger than that of the uplift in the windward side. For the ocean area, horizontal wind crossing the inclined boundary layer top is responsible for the ABL water vapour output, especially in the nearshore region. We admit the current analysis is preliminary, but it does indicate the characteristics of vertical exchange flux distribution with topography, and the significance of the interaction between mountain/sea and synoptic airflow. The topographic-dependent feature of water vapour vertical exchange should also be of general meaning to other complex terrain regions around the world."

**Minor comments**

*[R1.3] L60 - should be cancelling (not canceling).*

**Response:** Corrected.

*[R1.4] L80 - Boutle et al (2011, QJ, doi:10.1002/qj.783) is possibly a better reference here.*

**Response:** Accepted. This reference has been added in the revised manuscript at L88 and L665-667.

*[R1.5] L125 - would be better to refer to "grid length" than resolution, since that is what is quoted (the resolution is ~5 times the grid-length!).*

**Response:** Accepted. It has been corrected in the revised manuscript at L133.

*[R1.6] L225 - "variables were extracted from the vertical level closest to the top of the ABL" - why not interpolate the variables to the ABL height, using the levels either side. I'm slightly worried that choosing the closest level may give significant differences if that level is above or below the BL top, i.e. are you always choosing free tropospheric values or always choosing ABL values, or a random mix of the two? And does this matter?*

**Response:** In the calculation process of this study, the values at the upper model level closest to the ABL top were chosen. To discuss the effect of the values selection at the ABL top, a comparative calculation of water vapour vertical exchange flux is conducted, using the variables interpolated linearly to the ABL height from the levels on either side. Taking January, April, July and October 2018 as examples, the two sets of vertical exchange fluxes were compared by the Wilcoxon signed-rank test (Wilcoxon, 1945), a method widely used in paired samples difference test. The statistical results show that there was no significant difference between the two groups of exchange fluxes ($P<0.05$), with Cohen's d ranging from 0.003-0.025 (a standardized effect size for measuring the difference between two datasets, with a value below 0.20 meaning a very small effect). As shown in Fig. R3, the hourly evolutions of these two flux results are highly consistent and their values are very close during four seasonal representative months, with the correlation coefficients exceeding 0.99 and normalized mean errors below 5%. The above comparisons demonstrate that the values chosen in this study have no significant impact on the accuracy of the calculated vertical exchange fluxes of water vapour.

[Figure]

Figure R3. Hourly evolutions of two sets of calculation results of water vapour vertical exchange fluxes (left) and their scatter distributions (right) in (a) January, (b) April, (c) July, and (d) October 2018. The black and red lines represent the fluxes calculated using the upper layer values and interpolated values of the ABL top, respectively.

**References**

Gao, Y., Wang, H. J., and Chen, D: Precipitation anomalies in the Pan-Asian monsoon region during El Niño decaying summer 2016, International Journal of Climatology, 38, 3618–3632, doi:10.1002/joc.5522, 2018.

Wilcoxon, F: Individual comparisons by ranking methods, Biometrics Bulletin. 1 (6): 80–83, doi:10.2307/3001968. hdl:10338.dmlcz/135688,1945.

Xue, F., Zeng, Q. C., Huang, R. H., Li, C. Y., Lu, R. Y., Zhou, T. J.: Recent Advances in Monsoon Studies in China, Advances in Atmospheric Sciences, 32, 206-229, doi:10.1007/s00376-014-0015-8, 2015.

Zhou, W., Chen, W., and Wang, D. X.: The implications of El Niño-Southern Oscillation signal for South China monsoon climate, Aquatic Ecosystem Health & Management, 15, 14-19, doi:10.1080/14634988.2012.652050, 2012.

---

## Author Comment (AC2)

**Manuscript:** Water Vapour Exchange between Atmospheric Boundary Layer and Free Troposphere over eastern China: Seasonal Characteristics and ENSO Anomaly (egusphere-2023-1639)

**Response to Reviewer #2:**

**Summary**

*The manuscript discusses the water vapour exchange between the atmospheric boundary layer (ABL) and free troposphere (FT). The water vapour exchange between the ABL and FT is an important phenomenon related to e.g. precipitation, clouds, tropical cyclone formation etc. Therefore, it is quite important to improve the understanding of the water vapour exchange. The authors are using WRF simulations for seven years to study the phenomena. The model and the parameterization schemes used are well evaluated against meteorological observations. The manuscript is very well written and structured and, in my opinion, it is quite easy for the reader to follow. The structure and results are already quite good, and the methods used are well evaluated, the results are discussed and compared well to existing literature and also the uncertainties of the results are discussed well. Therefore, I have only minor suggestions before I can suggest the publishing of the paper.*

**Response:** We thank the reviewer for the positive evaluation of this manuscript. The response to each comment is listed below. The original comments are in *blue and italic*, our replies are in normal font. Bracketed numbers are used for referee comments (e.g., *[R2.1]*).

**Minor comments**

*[R2.1] Figure 2: Do you have an explanation why the model seems to be underestimating the ABLH in winter months, but overestimating during the summer months?*

**Response:** We infer that the model biases of ABLH are linked to the simulated temperature, which is underestimated in wintertime and overestimated during summertime (Table S1 and S2), thus leading to the lower winter boundary layer and the higher summer boundary layer. The simulation of these two variables (ABLH and temperature) involves many factors such as surface-atmosphere exchange, boundary layer turbulence, long and short wave radiation, cloud process and their interaction. Previous studies have pointed out that even with the same set of parameterization schemes, various model performances may be given in different seasons (Vautard et al.,

2012; Brunner et al., 2015). In the present study, there is insufficient observational data to verify these processes, and the analysis of the specific causes of model errors is beyond the research scope. Though for these biases in ABLH and other meteorological variables, major results and conclusions in this paper should not be altered.

**[R2.2]** *Figure 2 (+others): It would be good for the reader to point out in the caption, that the winter panel and the summer panels have different scales in y-axes.*

**Response:** Thanks for this suggestion. The scale differences have been pointed out in the captions of Fig. 2, Fig. 5, and Fig. 7.

**[R2.3]** *P8 L239: Please give proper citation for the ECMWF data used in the study.*

**Response:** Accepted. We have standardized the citation format for this dataset according to the journal requirements in the revised manuscript at L249-250, and provided the creators, title, repository, DOI and publication year in the references section.

**[R2.4]** *P9 L273: Do you mean Sect 3.2 instead of Sect. 3b?*

**Response:** Yes. This mistake has been corrected in the revised manuscript at L290.

**[R2.5]** *Figure 8: Is the map showing only statistically significant grids? If not what percentage of the grids were significant? Were there any spatial variation of the significancy?*

**Response:** Figure 8 shows all grids, not only statistically significant ones. In the whole research domain (20-42°N, 108-122°E), approximately 64% of grids are significantly correlated, with a confidence level of 95%. In terms of spatial variation, a positive-negative-positive triple distribution is presented in the correlation map. The proportion of significant grids is highest (~70%) in the central region (28-35°N, 108-122°E), followed by the southern area (~65%) and the northern area (~55%). This means that the central region has the most sensitive response to ENSO.

In the revised manuscript, the significant grids are indicated by black dots in new Fig. 8, and their percentage and spatial variation are supplemented at L457-464. These revisions are displayed below.

"The statistical result shows that there is a significant correlation between the two factors, with about 65% of the grids meeting the 95% confidence level. A positive-negative-positive triple distribution is presented in the correlation map (Fig. 8). On this basis, the sensitive areas are identified, in which the water vapour exchange fluxes are further analysed. The central region (28-35°N, 108-122°E) has the most obvious significance, where the proportion of significant grids is as high as 70%. This area shows a negative correlation, i.e., the mean vertical output flux of water vapour is enhanced by about 57.6~151.2 g m$^{-2}$ h$^{-1}$ in cold phase La Niña years, and vice versa in

warm phase El Niño years. In south (20-28°N, 108-122°E) and north (35-42°N, 108-122°E) areas with positive correlation coefficients, the trend is reversed. That is, the ABL moisture ventilation flux weakens 79.2~140.4 g m$^{-2}$ h$^{-1}$ in La Niña years and increases 108~194 g m$^{-2}$ h$^{-1}$ in El Niño years."

[Figure]

Figure R1. Spatial distribution of correlation coefficient between the water vapour exchange flux anomalies and Niño-3.4 index in July for 7 years. The dots indicate statistically significant grids and the black dashed lines indicate the triple distribution.

*[R2.6] P16 L458–459: Which section do you refer to with Sect. 3a?*

**Response:** It refers to Sect. 3.1.2. We are sorry for this mistake, and it has been corrected in the revised manuscript at L472.

*[R2.7] Summary: Even if it is good to have some sort of summary of the results, I would prefer (also or instead of summary) a short conclusions section that would also point out the most important findings of this study. In addition, it should be also clearly pointed out in the abstract.*

**Response:** Thanks for this suggestion. Reviewer #1 also gives a similar comment. In the revised manuscript, we have removed the detailed summary statements, replacing them with brief conclusions. This section is renamed as Conclusions and highlights the most important findings of this study. The abstract is also rephrased.

**References**

Brunner, D., Savage, N., Jorba, O., Eder, B., Giordano, L., Badia, A., Balzarini, A., Baró, R., Bianconi, R., Chemel, C., Curci, G., Forkel, R., Jiménez-Guerrero, P., Hirtl, M., Hodzic, A., Honzak, L., Im, U., Knote, C., Makar, P., Manders-Groot, A., van Meijgaard, E., Neal, L., Pérez, J. L., Pirovano, G., San Jose, R., Schröder,

W., Sokhi, R. S., Syrakov, D., Torian, A., Tuccella, P., Werhahn, J., Wolke, R., Yahya, K., Zabkar, R., Zhang, Y., Hogrefe, C., and Galmarini, S.: Comparative analysis of meteorological performance of coupled chemistry-meteorology models in the context of AQMEII phase 2, Atmospheric Environment, 115, 470-498, doi: 10.1016/j.atmosenv.2014.12.032, 2015.

Vautard, R., Moran, M. D., Solazzo, E., Gilliam, R. C., Matthias, V., Bianconi, R., Chemel, C., Ferreira, J., Geyer, B., Hansen, A. B., Jericevic, A., Prank, M., Segers, A., Silver, J. D., Werhahn, J., Wolke, R., Rao, S. T., and Galmarini, S.: Evaluation of the meteorological forcing used for the Air Quality Model Evaluation International Initiative (AQMEII) air quality simulations, Atmospheric Environment, 53, 15-37, doi: 10.1016/j.atmosenv.2011.10.065, 2012.

---

## Author Response (AR2)

**Manuscript:** Water Vapour Exchange between Atmospheric Boundary Layer and Free Troposphere over eastern China: Seasonal Characteristics and ENSO Anomaly (egusphere-2023-1639)

**Response to Reviewer #1:**

*[R1.1] I thank the authors for their responses - the revised manuscript is now much better. I have only one final comment, regarding my original R1.6.*

*The response to my comment appears to suggest that the value selected for the BL top is the nearest level BELOW the BL height - is that correct? If so, I think that would be worth clarifying in the text, which still suggests that it is the closest level, regardless of whether this is above or below the BL top. I would also suggest mentioning in the text that you see no significant effect from interpolating values to the BL top, as you have shown in the review responses.*

**Response:** We thank the reviewer for the positive evaluation and have followed the advice. The values selected for the BL top are extracted from the nearest level above the BL height. We have added this clarification in the text and indicated that there is no significant effect from interpolating values to the BL top. For your convenience, the revision is listed below:

"Other variables at the ABL top were extracted from the nearest model level above the ABL height (there is no significant difference between these extracted values and those interpolated to the ABL top)."